# Emerging Biomarkers and Advanced Diagnostics in Chronic Kidney Disease: Early Detection Through Multi-Omics and AI

**DOI:** 10.3390/diagnostics15101225

**Published:** 2025-05-13

**Authors:** Sami Alobaidi

**Affiliations:** Department of Internal Medicine, University of Jeddah, Jeddah 21493, Saudi Arabia; salobaidi@uj.edu.sa

**Keywords:** chronic kidney disease, emerging biomarkers, genomics, proteomics, metabolomics, AI-driven diagnostics, CKD risk prediction, personalized nephrology

## Abstract

Chronic kidney disease (CKD) remains a significant global health burden, often diagnosed at advanced stages due to the limitations of traditional biomarkers such as serum creatinine and estimated glomerular filtration rate (eGFR). This review aims to critically evaluate recent advancements in novel biomarkers, multi-omics technologies, and artificial intelligence (AI)-driven diagnostic strategies, specifically addressing existing gaps in early CKD detection and personalized patient management. We specifically explore key advancements in CKD diagnostics, focusing on emerging biomarkers—including neutrophil gelatinase-associated lipocalin (NGAL), kidney injury molecule-1 (KIM-1), soluble urokinase plasminogen activator receptor (suPAR), and cystatin C—and their clinical applications. Additionally, multi-omics approaches integrating genomics, proteomics, metabolomics, and transcriptomics are reshaping disease classification and prognosis. Artificial intelligence (AI)-driven predictive models further enhance diagnostic accuracy, enabling real-time risk assessment and treatment optimization. Despite these innovations, challenges remain in biomarker standardization, large-scale validation, and integration into clinical practice. Future research should focus on refining multi-biomarker panels, improving assay standardization, and facilitating the clinical adoption of precision-driven diagnostics. By leveraging these advancements, CKD diagnostics can transition toward earlier intervention, individualized therapy, and improved patient outcomes.

## 1. Introduction

Chronic kidney disease (CKD) is a global health challenge, affecting approximately 13.4% of the population and contributing significantly to morbidity and healthcare costs [1,2]. A major concern is its silent progression, with symptoms often becoming evident only in advanced stages. This delayed diagnosis limits opportunities for early intervention, underscoring the urgent need for biomarkers that enable earlier and more precise detection [3,4].

Current CKD diagnosis primarily relies on serum creatinine and estimated glomerular filtration rate (eGFR). However, these measures have well-documented limitations. Serum creatinine may remain within normal limits despite significant renal damage, while eGFR estimates can be imprecise due to variations in muscle mass, age, and sex [5,6,7]. Consequently, there is a growing shift toward novel biomarkers that can detect renal injury earlier and with greater accuracy (Figure 1).

Emerging biomarkers such as neutrophil gelatinase-associated lipocalin (NGAL), cystatin C, and soluble suppression of tumorigenicity 2 (sST2) offer significant advantages over traditional methods. NGAL increases within hours of kidney injury, while cystatin C is less affected by muscle mass or metabolic fluctuations [5,7,8]. Additionally, sST2 has demonstrated utility in both cardiovascular and renal risk stratification, reinforcing the need to explore such emerging markers [8].

Beyond individual biomarkers, multi-omics approaches—including genomics, proteomics, and metabolomics—are transforming CKD diagnostics by revealing complex molecular signatures of early disease [5,9]. Novel candidates such as fibroblast growth factor-23 (FGF-23), soluble urokinase plasminogen activator receptor (suPAR), and Klotho have been linked to disease progression, endothelial dysfunction, and cardiovascular events [10,11]. While promising, these biomarkers require large-scale validation before widespread clinical adoption.

Urinary biomarkers are also gaining interest as non-invasive diagnostic tools. Kidney injury molecule-1 (KIM-1) and urinary NGAL allow for real-time assessment of kidney health, enabling earlier and more targeted interventions [1,10]. The integration of these biomarkers with artificial intelligence (AI)-driven models has the potential to enhance diagnostic precision and usher in an era of personalized nephrology [3,5].

Given the rapid advances in biomarker discovery, a systematic evaluation of their clinical relevance, reliability, and feasibility for routine implementation is essential. Distinct from previous reviews, this manuscript integrates emerging biomarkers with multi-omics and artificial intelligence-driven diagnostics, highlighting their collective strengths and clinical translation challenges. This review emphasizes their potential to enhance early detection, refine prognosis, and support personalized treatment strategies in CKD.

Literature for this review was retrieved systematically from PubMed, Scopus, Web of Science, and Google Scholar, focusing on original research and relevant reviews published within the past decade. Key search terms included ‘CKD’, ‘biomarkers’, ‘multi-omics’, and ‘artificial intelligence’. Studies were excluded if they were case reports or lacked significant clinical biomarker data.

## 2. Overview of Conventional CKD Diagnostics

### 2.1. Serum Creatinine and eGFR

Serum creatinine remains the most widely used biomarker for estimating glomerular filtration rate (eGFR), a key parameter in CKD diagnosis (see Table 1 for a summary of conventional biomarkers and their limitations). Its availability and cost-effectiveness make it a standard tool in nephrology practice. However, serum creatinine levels are influenced by extrarenal factors, including muscle mass, age, sex, and diet, leading to potential inaccuracies in individuals with low muscle mass or altered metabolism [12]. Consequently, creatinine-based eGFR equations may fail to detect early kidney damage or provide precise risk stratification, highlighting the need for alternative biomarkers and refined assessment tools [1].

### 2.2. Urinalysis and Urine Microscopy Exam

Urine examination is a fundamental component of CKD diagnostics, offering insights into hematuria, proteinuria, and urinary casts, which can indicate underlying renal pathology [13]. While dipstick testing allows for rapid screening, urine microscopy provides a more detailed evaluation of kidney injury. Red blood cell (RBC) casts strongly suggest glomerulonephritis, while white blood cell (WBC) casts indicate interstitial nephritis. Granular or muddy brown casts are often associated with acute tubular injury, whereas hyaline casts may be seen in prerenal azotemia [14]. When interpreted in a clinical context, urine microscopy remains an essential, cost-effective diagnostic tool in nephrological practice.

### 2.3. Proteinuria

Proteinuria is a hallmark of CKD diagnostics, serving as both a marker of glomerular damage and a predictor of disease progression [15]. The albumin-to-creatinine ratio (ACR) is widely used for CKD staging and monitoring, providing valuable insights into glomerular integrity and cardiovascular risk [16]. However, proteinuria levels can fluctuate due to hydration status, physical activity, and physiological variability, complicating interpretation [17].

Recent studies emphasize the strong association between proteinuria and cardiovascular risk, yet it remains underrecognized as a cardiovascular risk factor despite its established link to coronary artery disease, stroke, heart failure, and arrhythmias [18]. Additionally, proteinuria in early CKD significantly increases the risk of major adverse cardiovascular events (MACE), heart failure, and mortality, underscoring its importance in risk stratification beyond eGFR alone [19]. These findings highlight the necessity of routine proteinuria assessment in CKD management to improve cardiovascular risk prediction and therapeutic decision-making.

### 2.4. Renal Imaging

Renal ultrasound plays a critical role in CKD diagnostics by evaluating kidney morphology, structural abnormalities, and potential urinary obstructions. It is the most commonly used imaging modality due to its non-invasive nature, accessibility, and affordability [16]. It is particularly useful for detecting cystic kidney diseases, hydronephrosis, and renal atrophy, which provide essential clues about underlying CKD etiology.

Recent advancements in ultrasound technology have further improved early CKD detection. Contrast-enhanced ultrasound (CEUS) enhances the visualization of renal perfusion, facilitating the detection of vascular abnormalities and microvascular dysfunction. Additionally, shear wave elastography (SWE) measures renal stiffness, correlating with fibrosis and CKD progression [20]. These techniques hold promise for enhanced early diagnosis and risk stratification, particularly in subclinical CKD cases.

### 2.5. Limitations and Emerging Perspectives

Despite their widespread use, conventional CKD diagnostic tools have inherent limitations. Serum creatinine and eGFR equations lack sensitivity for detecting early CKD, proteinuria levels are highly variable, and renal imaging largely provides anatomical rather than functional information. These shortcomings highlight the need for complementary biomarkers that offer earlier and more precise indicators of kidney dysfunction.

Among the emerging candidates, cystatin C, neutrophil gelatinase-associated lipocalin (NGAL), and kidney injury molecule-1 (KIM-1) have shown potential but require further validation for routine clinical use [21]. Additionally, AI-driven predictive models and multi-omics methodologies are being developed to enhance risk stratification and improve diagnostic accuracy, supporting a more integrative and personalized approach to CKD management.

Although serum creatinine, proteinuria, and renal imaging remain essential tools in CKD diagnosis, their limitations emphasize the need for complementary approaches. The integration of emerging biomarkers, AI-driven predictive models, and multi-omics strategies has the potential to enhance early detection, improve prognostication, and personalize CKD management. Future research should prioritize refining risk stratification models, improving biomarker standardization, and validating novel diagnostic tools to ensure widespread clinical application.

## 3. Rationale for Novel Biomarkers and Advanced Diagnostics

Traditional diagnostic tools, such as serum creatinine and eGFR, often fail to detect kidney dysfunction in its early stages, delaying interventions that could slow disease progression [22,23]. The integration of molecular diagnostics, multi-omics biomarkers, and AI-driven models is improving early detection, risk assessment, and personalized CKD management [10,24].

### 3.1. Unmet Needs in Early Detection

The limitations of traditional biomarkers create diagnostic gaps, particularly in early CKD. Serum creatinine, while widely used, is influenced by non-renal factors such as muscle mass, age, and sex, which can lead to underestimation of kidney dysfunction. Many patients experience no notable changes in eGFR or albuminuria until significant nephron loss has already occurred, reinforcing the need for biomarkers with higher sensitivity and specificity [10,23].

NGAL and RBP-4 can identify tubular injury before traditional markers indicate functional decline [9]. Likewise, microRNA-451 has demonstrated strong sensitivity and specificity for early diabetic nephropathy detection, representing a non-invasive alternative to conventional tests [25]. Fibrosis-associated biomarkers such as Matrix Metalloproteinase-7 (MMP-7), Monocyte chemoattractant protein-1 (MCP-1), and Dickkopf-3 (DKK3) have been proposed as indicators of progressive CKD, particularly in high-risk patients [5,24].

### 3.2. Improved Risk Stratification

Accurate risk stratification is essential for effective CKD management and ESRD prevention. While proteinuria and albuminuria remain key indicators of CKD progression, they may not fully predict long-term disease trajectory, particularly in normoalbuminuric diabetic nephropathy [23].

Urinary IgG and serum CX3CL1 have demonstrated predictive value in diabetic nephropathy progression, facilitating earlier identification of at-risk patients [26]. Similarly, inflammatory markers such as TNF-α, examined in the CRIC study, have been linked to CKD progression and cardiovascular risk [27]. Additionally, FGF-23 and suPAR are emerging as robust indicators of CKD progression, particularly in patients with coexisting cardiovascular conditions [10].

### 3.3. Potential for Personalized Medicine

Multi-omics advancements are reshaping precision nephrology, enabling tailored CKD management strategies. By integrating genomic, proteomic, and metabolomic data, researchers can refine treatment approaches based on individual disease profiles [28].

Among emerging biomarkers, cystatin C has demonstrated superior accuracy for estimating glomerular filtration, making it a valuable tool for personalized treatment adjustments [15]. TIMP-2 and IGFBP7 have been shown to guide early interventions aimed at preventing CKD progression. By identifying patient-specific molecular signatures, clinicians can more accurately predict disease progression and therapeutic responses. For instance, elevated cystatin C can guide clinicians in initiating earlier interventions to prevent rapid progression [15], while markers such as TIMP-2 and IGFBP7 help identify patients who might benefit from targeted preventive strategies against acute kidney injury (AKI) and subsequent CKD progression [29].

### 3.4. Applications in Prevention and Monitoring

Beyond diagnosis, novel biomarkers are increasingly used to monitor CKD progression and assess treatment responses. Tissue inhibitor of metalloproteinases-2 (TIMP-2) and Insulin-like growth factor binding protein-7 (IGFBP7), as highlighted by Engelman et al. (2019), have been associated with reducing AKI incidence following cardiac surgery, demonstrating the clinical utility of biomarker-guided interventions [30]. Similarly, urinary Procollagen type III amino-terminal propeptide (PIIINP) has been identified as a non-invasive marker of renal fibrosis, allowing for disease progression tracking and therapy response evaluation [31].

These biomarkers enable early detection of subtle kidney function changes, allowing for timely therapeutic adjustments and improved patient outcomes. Their application in routine monitoring enhances risk stratification models, leading to more effective long-term CKD management.

### 3.5. Artificial Intelligence (AI) and CKD Diagnostics

The integration of AI in CKD diagnostics is revolutionizing predictive modeling and real-time risk assessment. AI-driven tools have demonstrated superior diagnostic accuracy, outperforming conventional methods in forecasting CKD progression [32]. Additionally, AI-enhanced electronic health records (EHRs) enable automated identification of high-risk CKD patients, facilitating earlier interventions and improved clinical decision-making [33].

The combination of novel biomarkers, multi-omics diagnostics, and AI-driven models is reshaping CKD management. These advancements hold the potential to enhance early detection, refine risk stratification, and enable personalized treatment, ultimately improving patient outcomes. Future research should focus on standardizing biomarker applications, validating multi-center clinical data, and incorporating advanced diagnostics into routine nephrology practice to further strengthen CKD prevention and treatment strategies.

## 4. Emerging Biomarkers in CKD

The biomarkers selected for inclusion in this review were chosen based on robust clinical evidence, demonstrated diagnostic utility, and potential applicability for clinical integration. While most biomarkers discussed have relevance across diverse CKD etiologies, we explicitly highlight instances where certain biomarkers offer particular utility within specific etiologies (e.g., diabetic nephropathy) or stages of CKD (early detection vs. advanced disease monitoring). Recognizing CKD’s heterogeneous nature, our assessment underscores the need for further validation studies to determine the optimal context for clinical use.

### 4.1. Neutrophil Gelatinase-Associated Lipocalin (NGAL)

NGAL is a key biomarker for kidney injury, released from renal tubular cells, neutrophils, and other epithelial cells in response to tissue damage. It has demonstrated utility in both AKI and CKD, offering advantages over serum creatinine due to its early response to renal insult [34,35]. Its role extends beyond renal injury, as it is also involved in systemic inflammatory conditions.

#### 4.1.1. Diagnostic and Prognostic Utility in AKI

NGAL has strong diagnostic and prognostic value in AKI, outperforming serum creatinine in early detection. Byeon et al. (2022) demonstrated that NGAL effectively predicts CI-AKI in patients undergoing percutaneous coronary interventions, correlating with long-term adverse cardiovascular outcomes [36]. Engström et al. (2024) found that plasma NGAL independently predicts dialysis need and 90-day mortality in critically ill COVID-19 patients, highlighting its role in risk stratification [34]. These findings support NGAL’s role as an early biomarker of renal stress, reflecting both injury and systemic inflammation.

#### 4.1.2. Role in CKD Progression and Risk Stratification

Plasma NGAL has been associated with new-onset CKD in a general population cohort, identifying at-risk individuals before significant eGFR decline [37]. Serum and urinary NGAL correlate with eGFR and independently predict CKD progression and cardiovascular complications, underscoring its significance in comprehensive CKD management [35,38].

#### 4.1.3. Implications in Diabetic Kidney Disease (DKD)

NGAL has shown particular value in detecting early-stage DKD, especially in normoalbuminuric patients, a group often overlooked by traditional diagnostic methods. Studies have demonstrated that NGAL exhibits high sensitivity and specificity in identifying DKD, making it a non-invasive alternative to conventional diagnostics. Its expression is closely linked to inflammation and renal fibrosis, both of which play a pivotal role in DKD pathogenesis. Consequently, NGAL serves as a predictive marker for CKD progression in diabetic populations [35,39].

#### 4.1.4. Applications in Personalized Medicine

NGAL not only reflects kidney injury but may also be involved in renal repair mechanisms, suggesting a dual role in injury detection and recovery monitoring. Its predictive capacity for CKD progression, particularly in high-risk groups such as diabetic and hypertensive patients, positions it as a promising tool for personalized nephrology. Future research should focus on standardizing NGAL assays and integrating them into routine nephrology practice to enhance early CKD detection, risk stratification, and personalized treatment strategies [35,37].

### 4.2. Cystatin C

Cystatin C offers greater sensitivity in detecting early kidney dysfunction compared to creatinine, particularly in individuals with low muscle mass or variable dietary protein intake [40]. Its superior accuracy in GFR estimation makes it particularly beneficial for patients with diabetes or sarcopenia [41]. Unlike creatinine-based equations, which historically included race-based adjustments, cystatin C provides consistent results across racial and ethnic groups, eliminating potential biases [40]. Studies have demonstrated that cystatin C-based eGFR better stratifies cardiovascular risk in CKD patients compared to creatinine-based eGFR, emphasizing its clinical relevance [42].

#### 4.2.1. Integration into Clinical Practice and Foundational Evidence

Early research established cystatin C as a reliable marker of GFR, demonstrating higher sensitivity in detecting kidney dysfunction compared to creatinine [43]. These foundational studies provided the basis for its clinical adoption, particularly in CKD staging and risk stratification.

Research highlights its ability to detect early nephropathy, particularly in normoalbuminuric diabetic patients, where creatinine-based eGFR may underestimate renal impairment [44]. Additionally, the dual-marker equation (eGFRcr-cys), which combines cystatin C and creatinine, enhances CKD detection and stratification by leveraging both markers’ strengths [41].

#### 4.2.2. Applications in Specific Conditions

Cystatin C has shown strong correlations with microvascular complications in diabetic patients. Studies have demonstrated that discrepancies between cystatin C- and creatinine-based eGFR are linked to an increased risk of diabetic microvascular complications, highlighting its role in DKD risk assessment [44]. Additionally, cystatin C-based eGFR has been identified as a superior predictor of long-term mortality in critically ill patients, reinforcing its prognostic value in high-risk populations [45].

#### 4.2.3. Clinical Implications of Cystatin C

The integration of cystatin C into routine nephrology practice aligns with precision medicine, facilitating tailored interventions and more accurate monitoring of CKD progression. Its superior predictive accuracy in both renal and cardiovascular outcomes supports its expanding role in risk assessment and treatment planning [41,42]. Given its advantages over creatinine, cystatin C is expected to become a central component of future nephrology guidelines and clinical decision-making [45].

### 4.3. Kidney Injury Molecule-1 (KIM-1)

KIM-1 is a transmembrane glycoprotein expressed in proximal tubular epithelial cells following kidney injury. Under normal conditions, its expression remains low or undetectable; however, in response to ischemic, nephrotoxic, or inflammatory insults, KIM-1 is markedly upregulated, making it a promising biomarker for tubular injury. Its diagnostic utility spans both AKI and CKD, offering advantages over traditional markers like serum creatinine.

#### 4.3.1. Diagnostic Utility in Acute Kidney Injury (AKI)

KIM-1 is recognized as an early biomarker for AKI, demonstrating superior sensitivity compared to serum creatinine. A systematic review and meta-analysis found urinary KIM-1 (uKIM-1) significantly elevated in early-stage AKI, with a pooled area under the curve (AUC) of 0.93, underscoring its diagnostic accuracy [46]. Network meta-analysis results further reinforced its diagnostic value across diverse clinical settings [47].

Recent findings also emphasize KIM-1’s predictive role in post-surgical AKI. Studies have shown that plasma KIM-1 levels correlate with AKI severity, allowing earlier detection of tubular injury before elevations in serum creatinine and blood urea nitrogen (BUN) [48]. Furthermore, research has linked elevated KIM-1 levels to increased AKI risk following cardiac surgery, emphasizing its importance in high-risk surgical populations [49].

#### 4.3.2. Prognostic Value in CKD

KIM-1 has emerged as a robust predictor of CKD progression and adverse renal outcomes. Longitudinal studies have associated elevated plasma KIM-1 (p-KIM-1) levels with faster eGFR decline and increased CKD risk over 16 years, demonstrating its prognostic relevance in early-stage CKD [50]. Additionally, KIM-1 has been identified as a strong predictor of diabetic nephropathy progression, outperforming traditional CKD markers such as albuminuria and eGFR, particularly in normoalbuminuric patients [48].

#### 4.3.3. Applications in Specific Clinical Contexts

Beyond AKI and CKD diagnostics, KIM-1 is clinically significant in nephrotoxicity assessment. Research has demonstrated that elevated urinary KIM-1 levels precede clinically evident kidney injury in cisplatin-induced nephrotoxicity, enabling early dose adjustments and nephroprotective strategies [51]. Additionally, KIM-1 has been proposed as an early marker for DKD, as studies have shown that urinary KIM-1 levels are significantly elevated in normoalbuminuric diabetic patients, suggesting its potential in detecting subclinical tubular injury before albuminuria develops [52].

#### 4.3.4. Clinical Implications of KIM-1

The integration of KIM-1 into nephrology practice has significant potential for early detection of tubular injury, proactive interventions, and nephrotoxicity surveillance, particularly in oncology and transplant settings. Its association with CKD progression supports its role in identifying high-risk patients for targeted renal protection strategies.

Despite its advantages, widespread clinical adoption of KIM-1 is limited by the lack of standardized cutoff values and the need for large-scale prospective validation. Future research should focus on establishing clinically relevant thresholds, optimizing KIM-1-based prediction models for diverse populations, and incorporating KIM-1 into multi-biomarker panels to enhance kidney disease diagnostics and risk assessment. As evidence continues to accumulate, KIM-1 is expected to play a growing role in personalized nephrology care, improving early detection, disease monitoring, and therapeutic decision-making.

### 4.4. Soluble Urokinase-Type Plasminogen Activator Receptor (suPAR)

suPAR has emerged as a key biomarker of systemic inflammation and kidney disease progression. It is the circulating form of urokinase plasminogen activator receptor (uPAR), which is expressed on immune and endothelial cells. Elevated suPAR levels have been strongly linked to kidney function decline, incident CKD, and increased risk of ESRD, particularly in high-risk populations such as those with diabetes, hypertension, and cardiovascular disease [3,53]. Growing evidence suggests that suPAR is not only a biomarker but may also play a direct pathogenic role in kidney disease by promoting inflammation, podocyte dysfunction, and fibrosis through interactions with αvβ3 integrins on podocytes [54,55].

#### 4.4.1. Diagnostic and Prognostic Value in CKD

The primary clinical utility of suPAR in nephrology lies in its ability to predict CKD onset and progression. Unlike traditional markers such as serum creatinine and albuminuria, suPAR can identify patients at risk for CKD before significant functional impairment occurs. Studies have demonstrated that suPAR levels are significantly elevated in individuals who later develop CKD, even among those with normal baseline kidney function, highlighting its potential for early detection [53]. Similarly, research has linked suPAR to accelerated eGFR decline and progression to ESRD, particularly in autosomal dominant polycystic kidney disease (ADPKD), reinforcing its predictive value in high-risk populations [56].

Beyond its diagnostic role, suPAR also serves as a prognostic marker for CKD progression. Higher suPAR levels have been correlated with faster CKD progression, particularly in patients with a reverse dipping nocturnal blood pressure pattern, a known cardiovascular risk factor in CKD [57]. These findings suggest that suPAR could enhance risk stratification models, enabling earlier intervention for patients at high risk of renal decline.

#### 4.4.2. suPAR in Inflammation and Kidney Pathophysiology

suPAR functions as a key mediator of systemic inflammation and immune activation, linking inflammatory processes to renal dysfunction. Research has shown that suPAR levels are elevated in both AKI and CKD, suggesting that it plays a role in bridging acute kidney injury to chronic disease progression [54]. Pathogenic mechanisms involve suPAR’s interaction with podocyte αvβ3 integrins, leading to podocyte dysfunction, proteinuria, and glomerular sclerosis, ultimately contributing to kidney fibrosis and disease progression [55].

Beyond nephrology, suPAR has also been implicated in endothelial dysfunction and increased cardiovascular risk among CKD patients, reinforcing its relevance for broader risk assessment beyond kidney disease [3]. These findings highlight suPAR’s multisystem impact, underscoring its role as a biomarker of systemic disease rather than just a marker of kidney dysfunction.

#### 4.4.3. suPAR in Multi-Biomarker Panels

While suPAR alone provides valuable prognostic information, its predictive accuracy improves when combined with other biomarkers. Studies have proposed that multi-biomarker panels incorporating suPAR, TNFR-2, and KIM-1 offer superior CKD risk prediction compared to single biomarkers, supporting a systems biology approach to patient stratification [58].

Recent advancements in biosensor technology have further enhanced the feasibility of suPAR measurement in clinical practice. Rapid and cost-effective quantification methods are now available, facilitating point-of-care suPAR testing in outpatient settings [59]. These innovations support suPAR’s potential as a real-time risk assessment tool, enabling more personalized nephrology approaches.

#### 4.4.4. Clinical Implications and Future Directions of suPAR

Given its predictive value for CKD progression, suPAR holds promise as an early risk stratification tool, particularly for identifying high-risk patients before conventional markers detect renal impairment. However, challenges remain, including the need for standardized cutoff values and further research into potential confounders such as acute inflammation and cardiovascular disease. Additionally, ensuring assay standardization across different platforms will be essential for its clinical implementation.

Future large-scale prospective studies are required to develop definitive guidelines for suPAR-based risk assessment in nephrology. As research continues to validate its role in kidney disease progression, suPAR is expected to become an integral part of personalized nephrology care, improving early detection, risk stratification, and treatment decision-making.

### 4.5. Other Promising Biomarkers

Advancements in nephrology have underscored the need for novel biomarkers that surpass the limitations of traditional markers like serum creatinine and albuminuria. Emerging biomarkers—including beta-2-microglobulin (B2M), beta-trace protein (BTP), fibrosis-related markers, and multi-biomarker panels—offer improved diagnostic precision, risk stratification, and personalized treatment strategies in CKD management [60,61,62].

#### 4.5.1. Beta-2-Microglobulin (B2M) and Beta-Trace Protein (BTP)

B2M and BTP have emerged as alternative markers for estimating GFR, particularly in populations where creatinine-based equations are less reliable. B2M is a low-molecular-weight protein freely filtered by the glomerulus and reabsorbed in the proximal tubules. Elevated B2M levels correlate with CKD severity and cardiovascular risk, particularly in patients with chronic inflammation [60].

Dual-marker equations incorporating B2M and BTP improve GFR estimation by reducing non-renal confounders such as muscle mass and diet-related variability [61]. Their role in CKD staging and risk stratification is increasingly supported by clinical evidence [63]. Despite these advantages, the lack of standardized assays limits clinical adoption. Further validation is needed to support their integration into nephrology practice.

#### 4.5.2. Normoalbuminuric Markers

Normoalbuminuric CKD, particularly in diabetic kidney disease (DKD), presents a diagnostic challenge, as renal dysfunction can occur despite normal albuminuria levels. Tubular injury markers such as retinol-binding protein (RBP) offer a potential solution by detecting early renal impairment before albuminuria develops.

RBP is a low-molecular-weight protein reabsorbed by the proximal tubules. In tubular dysfunction, impaired reabsorption leads to increased urinary RBP excretion, making it a sensitive marker of proximal tubular damage [64]. Studies indicate that RBP levels correlate with declining GFR in normoalbuminuric diabetic patients, supporting its role in early CKD detection [11].

Further research is needed to standardize RBP assays and determine clinical cutoff values to enhance their diagnostic accuracy across diverse populations.

#### 4.5.3. Fibrosis-Related Markers

Renal fibrosis drives CKD progression, leading to irreversible structural damage and functional decline. Fibrosis-related biomarkers allow early risk stratification and may help guide antifibrotic therapies. MCP-1, a pro-inflammatory cytokine, plays a role in monocyte recruitment and tubulointerstitial inflammation. Elevated urinary MCP-1 levels correlate with fibrosis severity and renal function decline [65,66].

PIIINP, a marker of extracellular matrix remodeling, has been linked to progressive renal fibrosis and CKD progression. Studies have shown that urinary PIIINP reflects fibrotic activity and predicts renal allograft dysfunction [67]. Research has further linked elevated PIIINP levels to an increased risk of renal function decline and post-transplant fibrosis, supporting its prognostic utility in CKD progression and transplant monitoring [68].

Despite their potential, the lack of standardized assays limits their clinical application. Large-scale validation is required before these markers can be integrated into routine nephrology practice.

#### 4.5.4. Novel Biomarker Combinations

The integration of multiple biomarkers into predictive models enhances CKD risk stratification. Multi-biomarker panels leverage complementary markers to improve diagnostic accuracy.

For example, a panel incorporating MCP-1, TNFR-1, and suPAR demonstrated superior CKD progression prediction compared to single biomarkers [62]. Additionally, combining NGAL with liver-type fatty acid-binding protein (L-FABP) has improved early detection of contrast-induced AKI, reinforcing the value of multi-marker strategies in acute kidney stress settings [69].

These findings suggest that a systems biology approach, integrating multiple biomarkers, may optimize CKD diagnostics and facilitate individualized nephrology care.

#### 4.5.5. Proenkephalin (PENK), Fibroblast Growth Factor-23 (FGF-23), and Dickkopf-3 (DKK3)

Several novel biomarkers, including PENK, FGF-23, and DKK3, have gained attention for their roles in CKD progression and prognosis. PENK, a peptide derived from proenkephalin, serves as a real-time indicator of GFR, offering a more accurate assessment of kidney function compared to creatinine. Unlike creatinine, PENK levels are not influenced by muscle mass or dietary protein intake, making it particularly valuable for elderly and malnourished populations. Recent studies have demonstrated that PENK strongly correlates with measured GFR, supporting its potential role as an early predictor of CKD progression [70].

FGF-23, a key regulator of phosphate metabolism, has been increasingly recognized as a predictor of CKD progression and cardiovascular morbidity. Elevated FGF-23 levels are associated with higher mortality rates and worsening renal function, underscoring its relevance in risk stratification and patient management [68].

Another emerging biomarker, DKK3, has been linked to renal tubular injury and fibrosis, with research suggesting that elevated urinary DKK3 levels serve as an early indicator of CKD progression. Studies have demonstrated its predictive value in diabetic nephropathy and hypertensive kidney disease, highlighting its potential for early intervention and risk assessment [62]. Its role in predicting tubular damage highlights its potential for integration into CKD risk models.

The integration of PENK, FGF-23, and DKK3 into CKD risk models could enhance early disease detection and personalized nephrology care. However, further research is needed to validate their clinical applications, establish standardized cutoff values, and determine their utility in guiding therapeutic interventions.

#### 4.5.6. Clinical Implications and Future Directions of Additional Biomarkers

The clinical adoption of emerging biomarkers presents an opportunity to enhance CKD diagnostics and refine treatment approaches. Biomarkers such as B2M and BTP provide alternatives for GFR estimation, while tubular injury markers like NGAL and RBP enable earlier detection of kidney dysfunction. Fibrosis-related markers such as MCP-1 and PIIINP offer insights into disease progression, supporting targeted interventions to preserve kidney function.

Future research should focus on standardizing biomarker thresholds, validating their clinical utility in large-scale cohorts, and incorporating them into decision-support tools. These biomarkers represent significant advances in CKD diagnostics, offering improved early detection, risk stratification, and personalized treatment strategies. However, immediate challenges such as assay standardization and validation in diverse populations remain essential considerations.

From a practical perspective, urine biomarkers—notably NGAL, KIM-1, MCP-1, and DKK3—offer non-invasive, rapid, and repeatable testing; among them, urinary NGAL and KIM-1 are currently the closest to routine clinical adoption for early detection and ongoing monitoring of kidney injury. Serum biomarkers such as cystatin C, suPAR, PENK, and FGF-23 provide systemic assessment and early identification of declining kidney function; cystatin C and suPAR already have substantial clinical validation and are well positioned for wider routine use, pending further assay standardization.

By grouping biomarkers this way, clinicians can quickly match sample type with practicality, supporting efficient integration of the most ready-for-practice tests into routine nephrology care. Table 2 summarizes these biomarkers’ primary functions, clinical utilities, and limitations, underscoring their integration potential into clinical practice.

## 5. Multi-Omics Approaches in CKD

Recent advancements in multi-omics technologies have transformed CKD research, enabling integrated disease diagnostics and management. Combining genomics, transcriptomics, proteomics, and metabolomics provides insights into disease mechanisms, risk stratification, and personalized therapies. Genomics identifies genetic predispositions; transcriptomics reveals gene regulatory changes; proteomics highlights critical proteins; and metabolomics maps biochemical pathways reflecting kidney function. This integration enhances disease classification and targeted interventions, making multi-omics vital for precision nephrology.

### 5.1. Genomics

Genomics has revolutionized CKD research by identifying genetic predispositions, gene-environment interactions, and potential therapeutic targets. Advances in genome-wide association studies (GWASs), whole-genome sequencing (WGS), and polygenic risk scores (PRSs) have significantly improved CKD diagnostics and risk stratification, paving the way for personalized nephrology.

#### 5.1.1. Genome-Wide Polygenic Score (GPS) and Genetic Risk Prediction

PRSs have emerged as a powerful tool for CKD risk prediction by integrating multiple genetic variants into a susceptibility model. Studies have demonstrated that PRS models incorporating APOL1 risk genotypes can predict CKD risk across diverse populations, offering a framework for individualized risk assessment and early intervention [71]. Additionally, research has highlighted PRS utility in high-risk individuals, supporting its role in precision nephrology [72].

#### 5.1.2. Genomic Insights into CKD Pathophysiology

GWASs have identified key genetic loci associated with CKD, including UMOD, APOL1, and NPHS1, which contribute to disease susceptibility and progression. These findings have refined CKD diagnostics and therapeutic approaches, demonstrating how genetic variations influence kidney function and disease trajectories [73]. Furthermore, WGS has been effective in diagnosing CKD of unknown etiology, identifying a genetic diagnosis in 25% of cases, thereby reinforcing its role in advancing nephrological care [74].

#### 5.1.3. Gene Polymorphisms and CKD Risk

Certain gene polymorphisms have been implicated in CKD development and progression. Research has linked TNF-α-308G/A promoter polymorphisms to diabetic nephropathy, suggesting their utility in early risk stratification [75]. Additionally, variations in SLC22A2 and SLC47A1 have been associated with CKD progression via altered tubular transport function, underscoring the role of gene-environment interactions in CKD pathophysiology [73].

#### 5.1.4. Integrative Multi-Omics Approaches

The integration of genomics, proteomics, metabolomics, and transcriptomics enhances CKD diagnostics and precision medicine. Studies have demonstrated that combining genomic, proteomic, and metabolomic data provides a multi-dimensional perspective on CKD pathophysiology, enabling the development of advanced diagnostic tools [58]. Additionally, the role of WGS in CKD patients with unknown etiology has been highlighted, supporting its potential for routine genetic screening in nephrology practice [74].

#### 5.1.5. Non-Invasive Genetic Biomarkers

Advancements in non-invasive genomic biomarkers have facilitated early CKD detection. Circulating microRNAs, such as miR-451, have been validated as liquid biopsy-based biomarkers, demonstrating potential for early-stage CKD diagnosis [76]. Similarly, cell-free DNA (cfDNA) assays have been explored as non-invasive alternatives to traditional biopsy-based diagnostics, improving early identification of CKD [73].

#### 5.1.6. Future Directions in CKD Genomics

Despite significant progress in CKD genomics, several challenges must be addressed to fully integrate these findings into clinical practice. One key priority is the validation of PRSs across diverse populations, ensuring their applicability in global CKD risk prediction models. Additionally, the incorporation of genomic data into electronic health records (EHRs) could enhance real-time risk assessment and personalized treatment strategies, bridging the gap between research discoveries and clinical decision-making.

Emerging technologies such as CRISPR-based gene editing hold promise for correcting genetic defects in inherited kidney disorders, yet their application in CKD therapy remains in its early stages. Expanding multi-ethnic genomic research is also essential to improve the generalizability of findings, particularly in underrepresented populations. As genomic research continues to evolve, integrating multi-omics data with clinical nephrology will be crucial in enhancing early CKD diagnosis, refining risk stratification, and paving the way for precision nephrology.

### 5.2. Transcriptomics & Epigenetics

Transcriptomics and epigenetics provide insights into CKD pathophysiology by revealing gene expression patterns and regulatory mechanisms contributing to disease progression. Transcriptomics identifies dysregulated pathways in renal inflammation and fibrosis, while epigenetics examines heritable modifications, such as DNA methylation and histone modifications, that influence CKD without altering genetic sequences. Together, these approaches enhance disease understanding, refine diagnostics, and support personalized therapies.

#### 5.2.1. Role of Transcriptomics and Epigenetics in CKD

Transcriptomics has identified dysregulated pathways involved in renal inflammation, fibrosis, and metabolic dysfunction, offering novel insights into CKD pathogenesis. Advances in single-cell RNA sequencing (scRNA-seq) allow high-resolution mapping of renal cell populations, refining our understanding of the kidney cell-specific contributions to disease progression [77].

Epigenetic modifications—including DNA methylation, histone modifications, and non-coding RNAs—regulate gene expression and influence CKD progression. Studies have identified DNA methylation changes at loci such as JAZF1, PELI1, and CHD2, linking them to kidney function decline [78]. Histone modifications affecting inflammation and fibrosis-related genes have also been explored as therapeutic targets, with histone deacetylase inhibitors showing promise in mitigating renal fibrosis [79].

#### 5.2.2. MicroRNA Profiling as Biomarkers

MicroRNAs regulate post-transcriptional gene expression and influence key biological processes in CKD, including inflammation, fibrosis, and apoptosis.

miRNA-451 exhibits high sensitivity and specificity in detecting early tubular injury in diabetic nephropathy, supporting its diagnostic potential [25]. Similarly, miRNA-152-3p has been linked to CKD progression, particularly in patients with declining eGFR [26].

Urinary miRNA panels have also been investigated in autosomal dominant polycystic kidney disease (ADPKD), where distinct miRNA signatures correlate with disease progression and CKD subtypes [80]. In addition, circulating miRNA signatures have been linked to renal dysfunction and systemic inflammation, reinforcing their role in CKD pathogenesis [81]. Despite their promise, miRNA-based assays require further standardization before widespread clinical implementation.

#### 5.2.3. Epigenetic Mechanisms and Their Clinical Implications

Epigenetic modifications regulate gene expression without altering the DNA sequence, impacting fibrosis, oxidative stress, and metabolic pathways in CKD.

DNA methylation at RASAL1 has been linked to renal fibrosis, contributing to fibroblast activation and extracellular matrix deposition [77]. Epigenome-wide association studies (EWAS) have identified methylation markers at JAZF1, PELI1, and CHD2, reinforcing their potential as predictive biomarkers [78].

Histone deacetylase inhibitors have demonstrated potential in reducing renal fibrosis and inflammation, highlighting their therapeutic promise [79]. However, further research is required to validate epigenetic biomarkers and standardize diagnostic assays.

#### 5.2.4. Integration into Multi-Omics Platforms

Integrating transcriptomics and epigenetics with genomics, proteomics, and metabolomics enhances CKD biomarker discovery and precision medicine. Combining gene expression data, DNA methylation profiles, and proteomic insights improves our understanding of CKD mechanisms and supports personalized treatment approaches.

Multi-omics integration enhances patient stratification and predictive modeling of CKD progression. Transcriptomic and proteomic data refine risk assessment models, while epigenetic and proteomic markers provide insights into the molecular drivers of CKD and potential therapeutic targets [82]. Additionally, linking transcriptomic and metabolomic data has uncovered metabolic alterations associated with renal dysfunction, improving CKD diagnostic and prognostic models.

Despite these advancements, harmonizing multi-omics data across platforms and ensuring reproducibility across populations remain key challenges. Future efforts should focus on optimizing computational frameworks for integrating diverse datasets into clinically actionable insights.

#### 5.2.5. Future Directions in Transcriptomics & Epigenetics

As transcriptomic and epigenetic research advances, their integration into clinical nephrology is expected to enhance CKD diagnostics, risk stratification, and treatment strategies.

Single-cell transcriptomics continues to refine early-stage kidney dysfunction detection and facilitate targeted interventions. Additionally, CRISPR-based epigenetic editing has emerged as a potential strategy to reverse pathogenic epigenetic modifications, including DNA methylation and histone changes linked to CKD progression. While still in early development, these approaches may offer novel therapeutic avenues.

Despite progress, challenges remain in translating transcriptomic and epigenetic discoveries into clinical nephrology. Standardization of biomarker assays and harmonization of multi-omics datasets are essential for ensuring reproducibility across diverse populations. Future research should prioritize developing standardized pipelines for transcriptomic and epigenetic data integration, validating these biomarkers in large CKD cohorts, and incorporating them into predictive models.

As multi-omics technologies evolve, their integration into precision nephrology will enable earlier detection, refined risk stratification, and more effective personalized interventions.

### 5.3. Proteomics

Proteomics, the large-scale study of proteins, is transforming CKD diagnostics by facilitating biomarker discovery, refining pathophysiological insights, and improving risk stratification. Unlike conventional markers such as eGFR and albuminuria, proteomic analyses provide greater sensitivity and specificity, enabling the detection of biochemical changes that precede clinical CKD manifestations [83,84]. Advancements in mass spectrometry-based proteomics have led to the identification of protein panels associated with CKD severity and progression, strengthening their role in precision nephrology.

#### 5.3.1. Urinary Proteomics in CKD Diagnosis

Urinary proteomic studies have identified protein signatures strongly linked to CKD progression. Among the most validated proteomic classifiers, CKD273—a panel of 273 urinary peptides—has shown high accuracy in detecting early-stage CKD and predicting disease progression, particularly in diabetic patients. The PRIORITY trial confirmed CKD273’s ability to identify high-risk individuals before albuminuria onset, supporting its potential for early intervention strategies [63].

Beyond CKD273, large-scale proteomic analyses have uncovered additional biomarkers, including B2MG, FETUA, VTDB, AMBP, and CERU, all of which are significantly associated with kidney function decline [84]. These findings reinforce urinary proteomics as a valuable tool for improving CKD diagnostics, monitoring disease progression, and enhancing individualized risk stratification.

#### 5.3.2. Technological Advances in Proteomics

Proteomic profiling in CKD has greatly benefited from advancements in mass spectrometry-based technologies, particularly capillary electrophoresis–mass spectrometry (CE-MS), which has been instrumental in identifying urinary peptides with diagnostic potential [83]. Integrating proteomics with metabolomics and genomics is further refining CKD pathophysiology insights, providing a comprehensive framework for personalized risk assessment [58].

In plasma-based proteomics, high-throughput platforms such as SomaScan and Olink have identified circulating proteins predictive of CKD progression, leading to the development of a 65-protein risk model with strong predictive accuracy (C-statistic = 0.86) [85]. The increasing availability of these platforms is expected to enhance clinical risk prediction and facilitate drug target discovery, advancing precision nephrology.

#### 5.3.3. Future Directions in Proteomics

Proteomics holds significant potential for early CKD detection, patient stratification, and treatment monitoring, but further development is required to fully integrate these findings into clinical practice.

A key challenge is the standardization of proteomic biomarkers to ensure reproducibility and facilitate translation into routine clinical workflows. Establishing consistent assay protocols and reference ranges will enhance clinical utility. Additionally, validating urinary proteomic classifiers across diverse CKD populations is critical for improving reliability, particularly in addressing variability related to age, sex, ethnicity, and comorbid conditions.

The integration of proteomic data into electronic health records (EHRs) could enable real-time risk stratification and support personalized treatment strategies. As CKD management evolves toward precision nephrology, proteomics will play a central role in multi-omics frameworks that enhance biomarker discovery and therapeutic targeting.

Expanding multi-omics integration, particularly by combining proteomics with transcriptomics and epigenetics, will refine CKD diagnostic models and improve early intervention strategies. By characterizing CKD pathophysiology at multiple molecular levels, this approach supports the development of multi-dimensional diagnostic tools with enhanced predictive power.

As high-throughput proteomic platforms continue to advance, their incorporation into multi-omics models is expected to drive the next generation of CKD diagnostics, facilitating individualized and precise patient care.

### 5.4. Metabolomics

Metabolomics, the large-scale study of small-molecule metabolites, provides a dynamic snapshot of biochemical processes disrupted in CKD. By analyzing metabolic pathways, metabolomics enables the identification of novel biomarkers with greater sensitivity and specificity than traditional markers like creatinine and proteinuria, improving early diagnosis and risk stratification [86,87]. Given the metabolic alterations associated with CKD—including amino acid dysregulation, lipid imbalances, and oxidative stress—metabolomic profiling offers valuable insights into disease progression and therapeutic targets.

#### 5.4.1. Emerging Metabolomic Biomarkers in CKD

Metabolomics has identified previously unrecognized metabolic perturbations linked to CKD progression. Amino acid metabolism plays a critical role, with branched-chain amino acids (BCAAs), tryptophan metabolites, and purine derivatives emerging as key indicators of renal dysfunction. Declining BCAA levels (valine, leucine, and isoleucine) are associated with muscle protein catabolism and impaired nitrogen balance, common features of CKD-related metabolic dysfunction. Additionally, dysregulation of the kynurenine pathway in tryptophan metabolism has been linked to systemic inflammation and glomerular injury, with elevated levels of xanthurenic acid, 2-aminobenzoic acid, and hydroxypicolinic acid [88].

Recent studies have further expanded the landscape of metabolomic biomarkers. Nuclear magnetic resonance (NMR)-based metabolomics has identified lipid metabolism alterations, including elevated very-low-density lipoprotein (VLDL) and reduced high-density lipoprotein (HDL), both associated with increased CKD risk [86]. Additionally, hydroxyasparagine, pseudouridine, and adenine have shown strong correlations with GFR decline and CKD progression, outperforming creatinine-based models in diagnostic accuracy [87,89]. Large cohort studies, including CRIC, ARIC, and AASK, have validated pseudouridine and homocitrulline as predictors of glomerular filtration decline, reinforcing their role in CKD risk assessment [90].

#### 5.4.2. Metabolomics in CKD Risk Stratification

Beyond biomarker discovery, metabolomics has refined CKD risk stratification, enhancing predictive models. Lipidomic markers, particularly VLDL, HDL, and triglycerides, have been integrated into risk assessment frameworks, demonstrating strong associations with CKD progression in large population studies [86].

Pseudouridine and homocitrulline have been validated as reliable indicators of renal function decline across multiple CKD cohorts, including CRIC, ARIC, and AASK [90]. Similarly, hydroxyasparagine and C-glycosyltryptophan have been identified as superior to conventional creatinine-based markers for assessing kidney function, further refining personalized risk assessment [87]. These findings underscore metabolomics’ potential to enhance CKD prediction models, allowing for earlier and more precise risk stratification.

#### 5.4.3. Integration of Metabolomics into Multi-Omics Frameworks

The integration of metabolomics with genomics, proteomics, and transcriptomics has significantly advanced biomarker discovery and clinical applicability. Combining metabolomic markers with CKD273—a well-established proteomic classifier—has improved predictive accuracy for CKD progression, particularly in diabetic nephropathy [88].

Multi-omics frameworks demonstrate that metabolomic data complement proteomic and genomic insights, enabling a systems-level understanding of CKD pathophysiology [63,88]. Additionally, spatial metabolomics using matrix-assisted laser desorption/ionization mass spectrometry (MALDI-MSI) has enabled precise localization of metabolic alterations in kidney tissues, correlating metabolite distributions with histopathological CKD features [89]. The integration of spatial and bulk metabolomics has strengthened targeted therapy development based on precise metabolic signatures.

#### 5.4.4. Future Directions and Clinical Implementation

Despite its potential, several challenges remain in translating metabolomics into clinical practice. Standardization and validation of metabolic biomarkers across diverse populations are critical priorities, requiring large-scale, multi-cohort studies to ensure reproducibility. Additionally, the high costs associated with mass spectrometry and NMR-based metabolomics pose barriers to widespread clinical adoption, necessitating the development of cost-effective analytical platforms.

Integrating metabolomics into clinical workflows will be essential to harness its full potential in CKD management. Personalized nephrology approaches based on individualized metabolic profiles hold promise for tailored therapeutic interventions, optimizing patient outcomes, and reducing CKD-related morbidity and mortality. Advances in AI-driven metabolomic analyses may further enhance the clinical utility of metabolic biomarkers, enabling real-time risk stratification and disease monitoring [89,91].

Metabolomics has revolutionized CKD research by providing early diagnostic insights, refining risk stratification, and guiding personalized interventions. The discovery of novel metabolic markers, including adenine, kynurenine derivatives, and pseudouridine, has expanded the understanding of CKD progression and potential therapeutic targets. Future priorities include clinical validation and integration into multi-omics frameworks to bridge the gap between metabolomic discoveries and precision nephrology.

The discovery of multi-omics biomarkers—including genetic variants, transcriptomic regulators, proteomic signatures, and metabolic markers—has significantly enhanced CKD diagnostics and risk stratification. A comprehensive summary of key biomarkers and their clinical relevance across these omics platforms is presented in Table 3.

## 6. Advances in Imaging-Based Diagnosis

Imaging plays a critical role in CKD diagnosis, offering insights into renal structure, function, and pathology beyond traditional biomarkers like eGFR and albuminuria. Recent advancements in functional imaging, ultrasound techniques, and AI-driven analysis have improved early detection, risk stratification, and precision diagnostics in nephrology. These innovations enhance the assessment of renal perfusion, fibrosis, and microvascular integrity, contributing to a more comprehensive understanding of CKD pathophysiology.

### 6.1. Functional and Structural Imaging in CKD

Functional imaging has gained increasing importance in nephrology, particularly arterial spin labeling (ASL) and intravoxel incoherent motion diffusion-weighted imaging (IVIM-DWI). ASL quantifies renal blood flow without contrast agents, making it suitable for CKD patients at risk of contrast-induced nephropathy. IVIM-DWI enhances renal assessment by differentiating structural abnormalities from functional changes, offering a non-invasive alternative to biopsy for detecting early microvascular damage [92].

Ultrasound remains a cornerstone of CKD assessment. Super-resolution ultrasound and contrast-enhanced ultrasound (CEUS) have improved the detection of microvascular changes and early-stage fibrosis. Shear wave elastography (SWE) and acoustic radiation force impulse (ARFI) ultrasound enable non-invasive renal stiffness assessment, correlating with interstitial fibrosis and disease progression [20].

The Mayo Imaging Classification (MIC) has been validated as a predictive tool for CKD progression in autosomal dominant polycystic kidney disease (ADPKD), providing effective risk stratification [93].

Radiomics and AI-driven models integrated with ultrasound imaging further enhance diagnostic precision by enabling quantitative imaging analysis, reducing reliance on subjective interpretation [20].

### 6.2. AI-Driven Imaging and Predictive Modeling

AI has transformed nephrology imaging, enhancing diagnostic precision, automating segmentation, and improving CKD risk stratification. AI-powered analysis of ultrasound, MRI, and CT scans detects CKD-related pathology earlier, reducing dependence on subjective interpretation. Studies demonstrate that deep convolutional neural networks (CNNs) can accurately detect renal fibrosis and glomerular pathology, supporting automated diagnostic workflows [91].

AI-assisted segmentation of renal CT images enables precise measurement of cortical volume and parenchymal thickness, which are critical for early CKD detection [94]. Additionally, 3D deep learning models, such as nnU-Net, have enhanced renal CT segmentation, improving risk assessment and disease monitoring.

Beyond traditional imaging, AI is being explored in novel CKD screening approaches. Ultra-wide-field (UWF) fundus imaging integrated into deep learning models (UWF-CKDS) has demonstrated accuracy in predicting CKD risk from retinal vascular changes, offering a non-invasive and scalable screening tool [95].

Machine learning models have also been developed to predict CKD progression and alert clinicians to acute kidney injury (AKI), enhancing early intervention strategies [96]. These AI-driven imaging advancements facilitate real-time risk assessment, personalized CKD management, and improved clinical decision-making.

### 6.3. Future Directions and Clinical Implications Imaging-Based Diagnostics

While advanced imaging and AI technologies are revolutionizing CKD diagnostics, several challenges remain in clinical adoption. Standardizing imaging protocols and validating AI models across multi-center studies are crucial for ensuring reproducibility and consistency. Additionally, the high costs and limited accessibility of advanced imaging techniques pose barriers to widespread implementation, particularly in resource-limited settings.

The future of CKD diagnostics lies in integrating multi-modal imaging with multi-omics data, creating a comprehensive disease profiling system for precision nephrology. AI applications have the potential to further enhance diagnostic accuracy, refine risk stratification, and facilitate early CKD detection.

However, ethical and regulatory challenges—including data privacy, algorithmic bias, and transparency in AI decision-making—must be addressed to ensure equitable implementation. The transition to AI-powered nephrology requires collaboration across clinical research, healthcare policy, and computational science to optimize precision diagnostics and personalized CKD management.

## 7. Digital Health and Artificial Intelligence in CKD

Digital health and AI are transforming CKD diagnosis, management, and risk stratification by integrating clinical, imaging, and biomarker data. AI-driven models optimize precision diagnostics and early interventions, promoting proactive disease management and personalized medicine.

### 7.1. AI-Driven Biomarker Integration

AI has significantly enhanced biomarker integration, improving predictive accuracy and risk stratification, particularly in diabetic kidney disease (DKD), through optimized diagnostic performance of proteomics-based biomarkers such as CKD273, TNFR1, and TNFR2 [97].

CKD273, a urinary peptidomic biomarker panel, leverages AI to analyze 273 urinary peptides, identifying high-risk individuals before CKD onset. This AI-assisted approach improves early detection and risk stratification, supporting proactive disease management [63]. Additionally, AI-integrated multi-biomarker panels incorporating TIMP2 and IGFBP7 have demonstrated strong predictive value in identifying AKI and its progression to CKD, advancing precision nephrology [98].

By combining multi-omics data, clinical markers, and electronic health records (EHRs), AI enhances individualized risk prediction and supports evidence-based clinical decision-making.

### 7.2. Telehealth and Remote Monitoring

AI-powered telehealth has revolutionized CKD care by enabling remote monitoring, real-time assessments, and personalized interventions, improving patient engagement, adherence, and clinical outcomes [99].

Wearable AI-driven biosensors allow continuous tracking of eGFR, urinary albumin, and electrolyte balance, facilitating early detection of CKD deterioration and timely therapeutic adjustments. Remote monitoring technologies reduce hospital visits and promote patient autonomy, supporting home-based disease management [7].

The integration of AI with remote monitoring shifts nephrology toward preventive care, empowering patients in disease management and improving adherence and outcomes.

### 7.3. Innovations in Mobile Health and Decision Support

AI-powered mobile health (mHealth) applications, such as the My Kidneys & Me (MK&M) platform, significantly improve patient activation, self-management, and adherence to treatment regimens [100].

Additionally, clinical decision-support systems (CDSS) integrated with mHealth platforms enhance data-driven, personalized nephrology care. AI-driven CDSS enables early risk assessment, treatment optimization, and automated alerts for disease progression, facilitating evidence-based nephrology practice [101].

These advancements in mHealth and CDSS are shifting nephrology toward a more patient-centered, efficient, and personalized approach to CKD management.

### 7.4. Challenges and Ethical Considerations

Despite its transformative potential, AI in CKD management faces challenges related to data privacy, algorithmic biases, and regulatory constraints. Ensuring equitable AI implementation requires rigorous validation and standardization of AI models across diverse populations. Global AI governance frameworks emphasize data security, transparency, accountability, interpretability, and bias mitigation to foster clinician and patient trust. Successful AI integration also requires interoperability with existing health systems, regulatory approvals, and physician training [33].

### 7.5. Future Directions and Clinical Implications of Digital Health & AI

The future of digital health and AI in CKD lies in multi-modal data integration—combining imaging biomarkers, genomics, proteomics, and metabolomics—for enhanced risk prediction and personalized treatment. AI-driven models will further improve diagnostic accuracy, supporting proactive disease management and early intervention. However, ensuring equitable implementation requires addressing model generalizability, standardization, and ethical considerations through interdisciplinary collaboration among nephrologists, AI researchers, and policymakers.

## 8. Clinical Implementation and Guidelines

Translating research into clinical practice is essential for improving CKD management. Integration of biomarkers, genomic tools, AI, and digital health solutions advances personalized nephrology. Yet, regulatory challenges, cost-effectiveness concerns, and standardization remain key barriers.

### 8.1. Biomarker Integration into Clinical Workflows

The implementation of biomarker-based CKD diagnostics in clinical practice requires rigorous validation and regulatory approval. Biomarkers such as NGAL, KIM-1, and DKK3 have demonstrated promise in detecting early renal damage and predicting progression risk, particularly in diabetic kidney disease (DKD) [1]. Advancements in multi-omics AI models have further enhanced risk stratification by integrating urinary biomarkers with electronic health records (EHRs), improving disease predictions [102].

Despite these advancements, regulatory hurdles persist. The KDIGO 2024 guidelines emphasize the need for biomarker standardization and clinical cutoffs to enhance reliability across diverse populations [103]. The Kidney Health Initiative (KHI) Roadmap underscores gaps in regulatory approval, highlighting the necessity for large-scale validation studies and harmonization of clinical protocols to facilitate widespread adoption [104].

Addressing these regulatory challenges is crucial for transitioning biomarkers from research to routine clinical practice, ensuring accessibility and clinical utility.

### 8.2. Genomic Tools and Personalized Medicine in CKD

The incorporation of genomic biomarkers such as APOL1 risk variants and UMOD polymorphisms has significantly improved CKD risk prediction and patient stratification [77]. GWASs and whole exome sequencing (WES) have identified high-risk populations, allowing for early interventions and advancing precision nephrology approaches.

However, clinical implementation challenges remain. The high cost of genetic testing and limited accessibility in resource-limited settings hinder widespread use. Ethical concerns, particularly regarding genetic data privacy and potential discrimination, necessitate robust policy frameworks to ensure equitable application in nephrology [28]. Additionally, the lack of standardized guidelines for genetic counseling and risk assessment complicates clinical decision-making.

To overcome these challenges, developing standardized clinical guidelines and ethical policies is essential for the safe and effective implementation of genomic tools in CKD management.

### 8.3. Economic Considerations and Cost-Effectiveness

The economic feasibility of multi-omics and AI-based CKD diagnostics is a major factor in determining their clinical adoption. Cost-effectiveness analyses suggest that early biomarker-based detection models could reduce CKD-related hospitalizations and dialysis costs by enabling earlier interventions. AI-driven diagnostic platforms have demonstrated potential in lowering healthcare expenditures by identifying high-risk individuals before disease progression [105].

Despite these benefits, high implementation costs and limited financial support mechanisms remain challenges. Many of these diagnostic tools require specialized laboratory equipment, AI-driven software, and trained personnel, making them less accessible in resource-limited settings. Policymakers and healthcare systems must compare AI-based and biomarker-driven diagnostics with traditional approaches in terms of both clinical efficacy and cost savings.

A primary barrier to clinical adoption is the absence of reimbursement models supporting AI-driven and biomarker-based tests. Studies emphasize that without clear financial incentives, healthcare institutions may hesitate to invest in these technologies [106]. To ensure economic viability, healthcare policymakers must develop reimbursement frameworks that integrate these technologies into routine CKD screening and management.

Additionally, further research should focus on large-scale comparative cost-effectiveness analyses to determine whether early CKD detection using AI and biomarkers leads to long-term cost savings by delaying or preventing dialysis and kidney transplantation. Assessing the scalability of AI-based nephrology solutions across different healthcare systems will also be crucial to ensuring cost-effective implementation strategies.

### 8.4. Barriers to Adoption and Future Directions

Despite growing evidence supporting biomarkers, genomic tools, and AI-driven diagnostics, several barriers hinder their widespread adoption in CKD management. A critical challenge is the lack of standardization in biomarker assays, which affects reproducibility and limits clinical utility. Regulatory agencies require harmonized validation protocols to ensure biomarker-based diagnostics yield consistent, reliable results across diverse populations [103]. The absence of universally accepted clinical cutoffs for many emerging biomarkers further complicates their integration into nephrology practice.

Beyond standardization, the integration of AI-based diagnostics into clinical workflows presents significant regulatory and ethical challenges. AI models must undergo rigorous validation to ensure unbiased, accurate predictions, particularly for populations underrepresented in training datasets. Concerns regarding algorithmic bias, data privacy, and AI decision interpretability have raised questions about the reliability and fairness of machine learning applications in nephrology [107].

The KHI Roadmap (2022) emphasizes the need for clear regulatory pathways for digital biomarkers and AI integration into clinical practice. It highlights gaps in current regulatory frameworks and underscores the importance of comprehensive guidelines for standardizing biomarker assays and validating AI models in nephrology [104]. Collaboration between researchers, clinicians, and policymakers will be essential to ensure transparency, accountability, and equitable access to AI-powered diagnostics.

Another challenge is the lack of interoperability between AI-powered diagnostic tools and existing electronic health records (EHRs). Seamless integration is essential for automated risk assessment and real-time clinical decision support, yet many healthcare systems lack the necessary infrastructure to accommodate AI-driven applications.

To address these barriers, future research should prioritize large-scale, multi-center validation studies assessing the clinical utility of biomarkers and AI models in diverse populations. Regulatory agencies should develop clear frameworks for standardizing biomarker assays and integrating AI diagnostics into nephrology practice. These guidelines should define best practices for model validation, ethical data use, and transparency in AI decision-making, fostering trust among clinicians and patients.

The future of CKD diagnostics will depend on interdisciplinary collaboration among nephrologists, AI researchers, policymakers, and regulatory bodies. By overcoming standardization challenges, regulatory hurdles, and technical limitations, the healthcare community can fully leverage biomarkers, AI, and digital health innovations to enable early detection, personalized treatment, and cost-effective CKD management.

## 9. Limitations and Controversies in CKD Diagnostics

Despite significant advancements in CKD diagnostics, several limitations and controversies hinder the widespread adoption of novel biomarkers and diagnostic tools. Variability in biomarker performance, lack of assay standardization, concerns about overdiagnosis, and cost-effectiveness remain major barriers to implementation. Addressing these challenges is essential for translating emerging diagnostics into routine clinical practice.

### 9.1. Heterogeneity in Biomarker Studies

A significant challenge in CKD diagnostics is the variability in biomarker performance across different populations and clinical settings. Studies show that biomarkers such as neutrophil gelatinase-associated lipocalin (NGAL) and cystatin C exhibit biological variability, raising concerns about their reliability in distinguishing AKI from CKD progression [17]. This heterogeneity stems from population-specific variations in biomarker expression, which may impact diagnostic accuracy and generalizability.

Additionally, studies have demonstrated substantial assay-related variability in soluble urokinase plasminogen activator receptor (suPAR) levels across independent CKD studies, highlighting how non-standardized analytical methods can lead to conflicting conclusions regarding biomarker utility [108]. Multi-center validation studies are needed to improve clinical applicability and minimize population-specific biases. Without rigorous validation and standardization, biomarker-based diagnostics may yield inconsistent results, limiting their broader clinical utility.

### 9.2. Lack of Standardization in Biomarker Measurements

The absence of standardized assay protocols remains a major barrier to the clinical implementation of novel biomarkers, including cystatin C, CKD273, and NGAL. Inconsistent measurement methods across laboratories contribute to diagnostic variability, increasing the risk of misclassification of CKD severity. Studies emphasize the need for harmonized methodologies for proteomics-based biomarkers to ensure their adoption into clinical workflows [58]. Similarly, differences in suPAR measurement techniques have led to discrepancies in CKD risk prediction, even within similar patient populations [108].

To address these inconsistencies, the KDIGO 2024 guidelines advocate for standardized diagnostic protocols and international calibration of biomarker assays to enhance reproducibility across diverse populations [103]. Harmonization of analytical methods and clinical cutoffs is crucial to reducing inter-laboratory variability and ensuring biomarker reliability. Collaborative efforts between regulatory agencies, research institutions, and industry stakeholders are essential for establishing global standards and improving diagnostic accuracy.

### 9.3. Overdiagnosis and Unnecessary Interventions

While advanced biomarkers have improved diagnostic sensitivity, concerns have emerged regarding overdiagnosis, particularly in individuals with borderline CKD or minimal risk of progression. Research indicates that over-reliance on biomarkers such as cystatin C may lead to overclassification of CKD, prompting unnecessary interventions and increasing patient anxiety [7]. This issue is particularly relevant in populations with low disease progression risk, where aggressive diagnostic strategies may result in overtreatment and increased healthcare costs.

The economic burden of unnecessary CKD screening has also been questioned, especially in low-risk populations. Studies suggest that mass screening programs may not be cost-effective unless targeted toward high-risk groups, such as individuals with diabetes and hypertension [109]. These concerns highlight the need for balanced diagnostic strategies that integrate biomarkers with clinical risk assessment tools to enhance specificity and reduce unnecessary interventions. Developing personalized diagnostic algorithms that consider individual risk factors may help optimize CKD management while avoiding overtreatment.

### 9.4. Economic Implications and Cost-Effectiveness

The cost-effectiveness of novel CKD biomarkers and screening strategies remains a subject of debate. While early biomarker-based detection may reduce CKD-related hospitalizations and dialysis costs, the economic feasibility of these advanced diagnostics is uncertain. Studies demonstrate that universal CKD screening using cumulative eGFR-based statistics could be cost-effective under specific conditions, particularly when combined with AI-driven risk stratification [110]. However, others argue that CKD screening is financially viable only in high-risk populations, such as individuals with diabetes and hypertension, but may not be justified for the general population [109].

The KDIGO 2024 guidelines emphasize the need for targeted screening approaches to optimize healthcare resource allocation, ensuring that diagnostic efforts focus on populations at the highest risk [103]. Comparative cost-effectiveness analyses between novel biomarker-based screening methods and traditional diagnostic approaches are necessary to justify large-scale adoption. Additionally, further research should focus on evaluating the long-term financial sustainability of multi-omics and AI-based CKD diagnostics and developing reimbursement models that support their integration into routine clinical practice.

## 10. Future Perspectives

The landscape of CKD diagnostics is evolving rapidly, driven by advances in multi-omics technologies, digital health, and AI. These innovations offer earlier detection, improved risk stratification, and personalized treatment strategies. However, translating these advancements into routine clinical practice requires addressing regulatory, economic, and ethical challenges while ensuring equitable implementation.

### 10.1. Integration of Genomic and Epigenetic Biomarkers

Advancements in GWAS, PRS, and WES have enhanced understanding of CKD genetics, facilitating early detection and individualized risk stratification [77]. Genomic markers, such as APOL1 risk variants and UMOD polymorphisms, identify high-risk individuals who may benefit from early interventions.

Beyond genetic predisposition, epigenetic modifications, including microRNAs and DNA methylation signatures, provide additional diagnostic precision by refining disease classification and improving predictive accuracy for CKD progression and treatment response [111]. However, challenges such as high costs, limited accessibility, and lack of standardization hinder widespread adoption. Standardized diagnostic protocols, expanded research in diverse populations, and improved affordability are necessary for clinical integration.

### 10.2. Digital Biomarkers and AI Innovations

AI-driven platforms are transforming CKD care by integrating multi-omics data and clinical biomarkers into predictive models for early diagnosis and disease monitoring [63]. Explainable AI (XAI) has further improved the interpretability of AI-generated predictions, ensuring clinicians can confidently apply AI-assisted risk stratification tools in nephrology [112].

Despite these advancements, concerns remain regarding AI governance, data privacy, and algorithmic fairness. The potential for bias in AI models underscores the need for robust validation studies and regulatory oversight to ensure equitable deployment across diverse populations [107]. AI-powered decision-support systems, when combined with real-time monitoring through EHRs, hold promise for optimizing CKD management, but their widespread adoption requires standardization, physician training, and regulatory approval [113].

### 10.3. Emerging Technologies and Smart Diagnostics

Innovative technologies such as smart toilets, wearable biosensors, and home-based diagnostic devices are revolutionizing early CKD detection and continuous monitoring [1]. Wearable biosensors now enable real-time measurement of kidney injury biomarkers, including KIM-1, NGAL, and DKK3, facilitating earlier interventions and proactive disease management.

Additionally, extracellular vesicle (EV) biomarkers are emerging as minimally invasive tools for assessing renal dysfunction, presenting new opportunities for precision nephrology [114]. Advances in point-of-care testing (POCT) devices and lab-on-a-chip technologies are improving access to portable CKD diagnostics, particularly in underserved populations [115]. However, challenges related to regulatory approval, affordability, and user adoption must be addressed to facilitate clinical integration.

### 10.4. Metabolomics and Multi-Omics Synergy

Metabolomics has emerged as a key tool in CKD research, identifying novel biomarkers linked to renal metabolism and fibrosis pathways [115]. By integrating metabolomics with proteomics, genomics, and transcriptomics, researchers can achieve a more comprehensive understanding of CKD progression, refining biomarker discovery and therapeutic targeting [116].

Despite its potential, high costs, complex analytical requirements, and the need for standardization remain barriers to implementation. Research should prioritize harmonizing multi-omics methodologies, improving data integration frameworks, and developing cost-effective solutions to ensure broader clinical adoption.

### 10.5. Age-Specific Research Needs and Health Disparities

Older adults account for a large proportion of CKD patients, yet many advanced biomarkers lack validation in elderly populations [117]. Differences in renal physiology, metabolic profiles, and biomarker expression patterns among older adults necessitate targeted research to improve diagnostic accuracy in this population.

Beyond age-related challenges, racial and socioeconomic disparities remain a significant concern in CKD diagnostics. Populations carrying genetic susceptibility markers, such as APOL1 risk variants, often have limited access to advanced diagnostic tools, exacerbating inequities in CKD outcomes [113]. Addressing these disparities requires expanding multi-ethnic validation studies, increasing accessibility to precision nephrology tools, and implementing policies that ensure equitable distribution of novel diagnostic technologies.

### 10.6. Long-Term Research Directions

Future research should prioritize large-scale, multi-center longitudinal studies to validate emerging biomarkers and diagnostic technologies across diverse populations. The rapid evolution of gene-editing technologies, such as CRISPR-based approaches, presents new opportunities for addressing CKD-related genetic mutations, potentially offering curative therapies [77].

Additionally, cost-effectiveness analyses will be essential for integrating novel diagnostic tools into routine clinical practice, ensuring that healthcare systems can sustainably implement precision nephrology solutions [110]. By fostering interdisciplinary collaboration, refining AI-driven models, and expanding precision medicine frameworks, CKD care can transition toward more personalized, predictive, and proactive approaches, ultimately improving patient outcomes and healthcare efficiency.

## 11. Conclusions

Emerging biomarkers and advanced diagnostic approaches, including multi-omics and AI, hold significant promise for transforming CKD management. These novel biomarkers offer enhanced sensitivity and specificity compared to traditional markers, allowing for earlier detection, precise risk stratification, and improved prediction of disease progression. Integrative multi-omics approaches, which combine genomics, proteomics, metabolomics, and transcriptomics, provide deeper insights into disease mechanisms and facilitate the move toward personalized CKD care. Moreover, AI-driven predictive models present powerful tools for real-time, individualized risk assessment. Despite these promising developments, substantial challenges remain, particularly regarding biomarker standardization, clinical validation, and practical integration into healthcare settings. Future research efforts must therefore prioritize refining biomarker panels, standardizing diagnostic methodologies, and establishing clear clinical guidelines to maximize the clinical impact of these innovations.

### Key Points

Emerging biomarkers such as NGAL, cystatin C, suPAR, TIMP-2, and IGFBP7 demonstrate superior sensitivity and specificity compared to traditional markers, enabling earlier CKD detection and improved patient stratification.Multi-omics approaches integrating genomics, proteomics, metabolomics, and transcriptomics significantly enhance diagnostic accuracy and facilitate personalized CKD management strategies.AI-driven predictive models offer powerful, real-time risk assessment tools, enabling earlier identification of high-risk CKD patients compared to conventional approaches.Significant hurdles remain regarding biomarker standardization, extensive clinical validation, and integration into routine clinical workflows, underscoring the necessity for ongoing research.Future research must prioritize refining and validating biomarker panels, standardizing assays, and clearly defining clinical guidelines to fully leverage these diagnostic advancements for improved CKD outcomes.

## Figures and Tables

**Figure 1 diagnostics-15-01225-f001:**
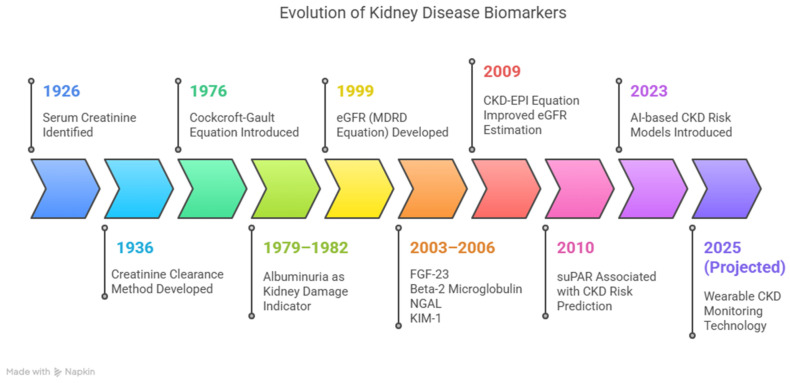
Evolution of kidney disease biomarkers (1926–2025). This timeline visually represents the historical development of CKD biomarkers, transitioning from conventional markers like creatinine and albuminuria to emerging tools such as NGAL, suPAR, and AI-driven risk models.

**Table 1 diagnostics-15-01225-t001:** Conventional CKD biomarkers and their limitations.

Biomarker	Strengths	Limitations
Serum Creatinine	Widely available, inexpensive	Late-stage detection, affected by muscle mass, diet, age, and sex
eGFR (Creatinine-based)	Standardized, globally used	Influenced by age, sex, muscle mass, and metabolic factors, reducing accuracy in certain populations
Proteinuria (ACR included)	Predicts CKD progression, cardiovascular risk	Highly variable due to hydration, physical activity, and transient conditions
Urinalysis	Detects hematuria and proteinuria	Detects abnormalities but cannot determine CKD cause or stage
Urine Microscopy	Identifies RBC, WBC, and casts for CKD assessment	Requires expertise for accurate interpretation, variability in readings
Renal Ultrasound	Identifies structural abnormalities, obstruction	Cannot assess early CKD or renal function directly
Advanced Renal Imaging (CEUS, SWE)	Improves fibrosis and microvascular damage detection	Limited availability, requires specialized training

**Table 2 diagnostics-15-01225-t002:** Emerging biomarkers in CKD—mechanisms, clinical applications, and limitations.

Biomarker	Mechanism	Clinical Applications	Limitations	Validation Status
NGAL	Released from tubular epithelial cells in response to injury	Early detection of AKI, CKD progression marker	Affected by systemic inflammation, not specific to kidney injury	Widely studied, available for clinical use for AKI
KIM-1	Upregulated in proximal tubular cells after injury	Sensitive marker for AKI, predictor of CKD progression	Requires standardization of assay cutoffs, limited availability in routine practice	Under validation, used in nephrotoxicity studies
suPAR	Inflammatory biomarker linked to podocyte dysfunction and fibrosis	Predicts CKD onset and progression, associated with cardiovascular risk	Influenced by systemic inflammation, lacks standardized cutoffs	Increasing clinical interest, studies support risk prediction
B2M	Freely filtered and reabsorbed in tubules; marker of filtration function	Alternative to creatinine-based eGFR, CKD severity indicator	Elevated in inflammatory conditions, not kidney-specific	Limited routine use, under evaluation
BTP	Low-molecular-weight protein used for GFR estimation	Enhances CKD staging, complements B2M for filtration assessment	Requires further validation for clinical adoption	Emerging, gaining interest for eGFR refinement
FGF-23	Regulator of phosphate homeostasis, linked to CKD-mineral bone disorder	Predictor of CKD progression and cardiovascular morbidity	Variability in assay results, influenced by dietary phosphate	Increasing use in research and risk assessment
DKK3	Tubular stress biomarker involved in fibrosis pathways	Early CKD detection, prognostic marker for renal fibrosis	Lacks standardized reference ranges	Early-stage validation, promising for risk stratification
PENK	Peptide reflecting real-time GFR	More accurate than creatinine in dynamic kidney function assessment	Limited commercial assays available	Research phase, high potential for precision nephrology
MCP-1	Chemokine involved in monocyte recruitment and renal inflammation	Indicator of fibrosis severity, CKD progression marker	Affected by systemic inflammatory diseases	Experimental, not yet widely adopted
PIIINP	Marker of extracellular matrix turnover and collagen deposition	Assesses fibrosis severity, predicts renal allograft dysfunction	Assay standardization needed, requires further validation	Research phase, being explored for transplant monitoring
Multi-Biomarker Panels	Integrates multiple biomarkers for improved CKD risk stratification	Enhances predictive accuracy by combining different pathophysiological pathways	Cost and accessibility limitations, lack of established guidelines	Gaining traction, panels under evaluation for clinical implementation

**Table 3 diagnostics-15-01225-t003:** Multi-omics biomarkers in CKD—mechanisms, clinical applications, and limitations.

Category	Biomarker	Function	Clinical Relevance	Validation Status
Genomics	APOL1 Genetic Variants	Associated with podocyte dysfunction and increased CKD risk, particularly in African ancestry populations	Genetic screening for early identification of high-risk individuals	Well-established, used in risk prediction models
	UMOD Gene Variants	Implicated in CKD progression via sodium transport dysregulation	Potential target for precision medicine in CKD	Under research, potential for therapeutic applications
	COL4A5 Variants (Alport Syndrome)	Causes inherited CKD via glomerular basement membrane defects	Genetic diagnosis crucial for early intervention in hereditary CKD	Clinically validated, used for hereditary CKD diagnosis
Transcriptomics & Epigenetics	miRNA-451	Biomarker for diabetic nephropathy, linked to tubular injury	Highly sensitive and specific for early-stage CKD detection	Early validation, requires large-scale cohort studies
	miRNA-152-3p	Predictor of CKD progression, associated with eGFR decline	Useful for monitoring disease progression in at-risk populations	Experimental, requires further validation
	DNA Methylation at RASAL1	Epigenetic marker of renal fibrosis	Potential therapeutic target for renal fibrosis interventions	Promising, under investigation for clinical translation
Proteomics	CKD273 Peptide Panel	Predictive panel for CKD progression and diabetic nephropathy	Validated in clinical trials (PRIORITY) for early detection and intervention	Well-validated, incorporated into multi-omics models
	Plasma Proteomic Markers (B2MG, FETUA, VTDB, AMBP, CERU)	Circulating proteins associated with CKD progression and kidney function decline	Identified in proteomic risk models but requires further validation in large cohorts	Experimental, undergoing validation in CKD cohorts
	Urinary Peptide Profiling (CE-MS-Based)	Mass spectrometry-based identification of urinary peptides for CKD risk assessment	Improves early-stage CKD detection and progression monitoring	Increasing clinical use, pending standardization
Metabolomics	Branched-Chain Amino Acids (BCAAs)	Dysregulation linked to muscle catabolism and metabolic imbalance	Potential metabolic intervention target for CKD progression	Under research, potential for clinical application
	Tryptophan Metabolites (Xanthurenic Acid, Hydroxypicolinic Acid)	Indicators of systemic inflammation and oxidative stress	Identifies high-risk CKD patients for early intervention	Emerging, requires further large-scale validation
	Adenine	Novel fibrosis marker, particularly in diabetic kidney disease	Early diagnostic and prognostic marker for CKD	Under investigation, promising for CKD monitoring
	Hydroxyasparagine	Marker for kidney function assessment superior to creatinine	Enhances pathway-specific biomarkers in personalized CKD risk assessment	Early-stage research, not yet standardized
	Pseudouridine & Homocitrulline	Indicators of glomerular filtration decline	Validated in CRIC, ARIC, and AASK cohorts for CKD risk stratification	Clinically validated in multiple CKD cohorts
	Lipid Metabolites (VLDL, HDL, Triglycerides)	Dysregulated lipid metabolism associated with CKD risk	Integrated into predictive models for enhanced CKD risk assessment	Increasingly recognized, requires clinical validation
Multi-Omics Integration	Multi-Biomarker Panels (e.g., suPAR + TNFR-1 + MCP-1)	Combines markers of inflammation, fibrosis, and kidney function	Validated in inflammation-associated CKD risk models; enhances predictive accuracy	Gaining traction, undergoing clinical evaluation

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
