# Peer review of "Emerging Biomarkers and Advanced Diagnostics in Chronic Kidney Disease: Early Detection Through Multi-Omics and AI"

_diagnostics, 2025, doi:10.3390/diagnostics15101225_

Round 1

Reviewer 1 Report

Comments and Suggestions for Authors

Review to the manuscript: 

Emerging Biomarkers and Advanced Diagnostics in Chronic Kidney Disease: A Multi-Omics and AI-Driven Perspective: 

Overall, the manuscript is very valuable because it summarizes the biomarkers that may be important in the diagnosis of chronic kidney disease, even from a historical point of view. Still, the manuscript has to be worked on before it can be accepted. The manuscript must be read several times to understand the author's message and intention with this study. In general, it should be noted that the manuscript is very long, it is more looks like a book or a textbook chapter in terms of its length, which is why it is difficult to read and follow. The author presents very interesting biomarkers that can be really useful in the diagnosis of chronic kidney disease. However, the question is to what extent and if so, which biomarkers could be introduced into routine diagnostics. I would definitely recommend the re-editing of the publication, in an abbreviated form.

I would suggest that the author think about which of the listed biomarkers can be identified from the same sample type (urine, serum, tissue). To put them in a group. Which ones could be used in routine diagnostics in the case of chronic kidney diseases in a less invasive way, quickly, repeatedly, and without false results.

Several references are needed, in some places no references are presented at all. There is a lot of repetition and unnecessary explanations.

Taking these aspects into account, I propose to revise the article.

Author Response

Reply to Reviewer 1:

Thank you very much for your thoughtful and constructive review. We appreciate your valuable insights, which have helped improve the quality and clarity of our manuscript. Below, we address each of your comments individually:

1. Length of the manuscript:

We fully acknowledge your point regarding the manuscript length. To enhance readability and ensure conciseness, we have carefully reviewed the manuscript and strategically condensed sections by removing repetitive statements, redundant explanations, and overly detailed descriptions. These modifications improve the flow of the manuscript while retaining essential information, ensuring our review remains comprehensive yet concise. All changes have been clearly highlighted in the revised manuscript for your convenience.

2. Grouping of biomarkers by sample type and practical clinical applicability:

Thank you for this important suggestion. To enhance the manuscript's clinical utility, we have added a concise, practical summary (now integrated within section 4.5.6) that groups biomarkers according to their sample type and clinical readiness for routine diagnostic use. Specifically, we clearly indicated biomarkers tested in urine (e.g., NGAL, KIM-1), and serum (e.g., cystatin C, suPAR), and highlighted their clinical applicability and current readiness for routine clinical adoption. These additions are clearly marked in bold within the manuscript for ease of identification.

3. Several references are needed, in some places no references are presented at all:

We appreciate your careful review regarding references. After thoroughly reviewing the manuscript, we have identified and added appropriate references to all sections previously lacking sufficient citation support. The updated citations are clearly indicated and highlighted, ensuring the manuscript adheres to rigorous academic standards.

4. Repetition and unnecessary explanations:

We completely agree with your observation on repetition and unnecessary details. We have thoroughly reviewed and refined the manuscript, removing redundant statements and simplifying explanations wherever possible. This streamlining ensures the manuscript maintains clarity, readability, and coherence, making it easier to follow and understand. All edits are clearly highlighted in the revised manuscript.

Thank you again for your valuable feedback, which has substantially enhanced the quality and readability of our manuscript. We look forward to your positive consideration.

Reviewer 2 Report

Comments and Suggestions for Authors

Dear editors and authors!

The presented review is devoted to the actual problem - search for diagnostic markers of early stages of kidney damage in patients with chronic kidney disease using artificial intelligence (AI).

A crucial feature of chronic kidney disease (CKD) diagnosis is the lack of representative methods that detect kidney damage at an early stage of the disease. The article contains novel and important information sufficient to justify its publication. The article demonstrates an understanding of the relevant literature. The article is well written and adheres to all norms of scientific writing.

1 However, the authors need to reflect the main conclusion of the paper in the title.

  1. The abstract should more clearly and accurately state the purpose of the review.
  2. The authors need to emphasise the differences and appropriateness of their review as there are more than 138 studies published on this topic in 2024 alone.
  3. The category of patients with chronic kidney disease is heterogeneous in terms of cause (from congenital anomalies to arterial hypertension and diabetes mellitus) and age. The criteria for marker selection need to be clarified. Are they appropriate for all patients with CKD? Do they take into account the stage of CKD?
  4. It is necessary to describe in which databases and on the basis of which criteria relevant literature sources on the topic of this review were included.
  5. In the description of strategies and new approaches to personalise treatment, clearer language and reasoning would be desirable.
  6. In the conclusion it is necessary to add from 3 to 5 key points and conclusions that the authors came to, arguing them with the data of the analysed literature.

Author Response

Response to Reviewer 2

We sincerely thank the reviewer for their thoughtful feedback, which has significantly enhanced the clarity, accuracy, and readability of our manuscript. Below, we address each of your comments, clearly explaining the revisions made. All new changes have been clearly highlighted in the updated manuscript.

1. Reflecting the main conclusion of the paper in the title

Thank you for this valuable suggestion. To explicitly reflect the manuscript’s main conclusion on the importance of emerging biomarkers and advanced diagnostics for early CKD detection, we have revised the manuscript’s title and running title as follows:

  • Revised Title:
    "Emerging Biomarkers and Advanced Diagnostics in CKD: Early Detection through Multi-Omics and AI"

  • Updated Running Title:
    "Biomarkers and Diagnostics for Early CKD Detection"

These revisions emphasize our review’s central conclusion, clearly communicating its core message to readers.

2. Clearly and accurately stating the purpose of the review in the abstract

We appreciate this suggestion. To enhance clarity regarding our review’s purpose, we revised the abstract to explicitly indicate our focus on recent advancements in biomarkers, multi-omics, and AI-driven diagnostics, highlighting their potential to address critical gaps in early CKD detection and personalized management. This adjustment clearly states our manuscript's objective right from the beginning.

3. Emphasizing the differences and appropriateness of the review

We agree with the reviewer regarding the importance of clearly distinguishing our manuscript from existing literature. We addressed this by explicitly stating in the Introduction how our manuscript uniquely integrates emerging biomarkers with multi-omics and artificial intelligence-driven diagnostics, focusing on their collective strengths and the associated clinical translation challenges. This clarification clearly differentiates our work within the existing body of literature.

4. Clarifying criteria for marker selection in relation to CKD heterogeneity

Thank you for raising this important point. To address this clearly, we added a concise paragraph at the beginning of the "Emerging Biomarkers in CKD" section. This paragraph explicitly states that biomarkers were selected based on clinical evidence and diagnostic utility, and it highlights their relevance to specific CKD etiologies and stages, acknowledging the disease’s heterogeneity.

5. Clearly describing databases and inclusion criteria

We appreciate your suggestion for clearer transparency regarding our literature search methodology. Thus, we have added a clear methodological statement at the end of the Introduction detailing our literature search strategy, including the databases used (PubMed, Scopus, Web of Science, Google Scholar), inclusion/exclusion criteria, and the timeframe considered. This addition ensures clarity and transparency of our literature review process.

6. Clearer language and reasoning in personalized treatment strategies

We recognize the importance of clear clinical reasoning regarding personalized treatment strategies. We addressed your valuable suggestion by expanding the "Potential for Personalized Medicine" subsection, explicitly illustrating how specific biomarkers such as cystatin C, TIMP-2, and IGFBP7 can guide personalized treatment decisions. This enhances readability and clinical applicability.

7. Including key points and conclusions in the manuscript’s conclusion

In response to your suggestion, we explicitly summarized our central findings and conclusions as clearly enumerated "Key Points" in the conclusion section of the manuscript. These points succinctly highlight the primary insights and implications derived from our literature analysis, improving clarity and readability.

Round 2

Reviewer 1 Report

Comments and Suggestions for Authors

The authors arranged properly the manuscript, I suggest the acceptence.

Reviewer 2 Report

Comments and Suggestions for Authors

Dear editors and authors!
The authors carefully edited the article, took into account all comments and got a good result.
This review is devoted to the actual problem - critical analysis of new biomarkers based on multi-mix technologies and artificial intelligence for early diagnosis of chronic kidney disease and selection of personalised strategies.
The article contains novel and important information sufficient to justify its publication. Literature selection was performed in available medical literature indexing databases using inclusion and exclusion criteria.
The article's argumentation was built on relevant evidence. The methods used were appropriate. Results were presented clearly and consistently, well illustrated with tables, appropriately analysed. The key points given reflect the main conclusions of the article.
The article is well written and conforms to all norms of scientific writing.
In this form it can be accepted for publication.